# An eight-season analysis of the teams' performance in the Spanish LaLiga according to the final league ranking

Ibai Errekagorri [1]*, Javier Fernandez-Navarro[2], Roberto López-Del Campo[3], Ricardo Resta[3], Julen Castellano[1]

1 Society, Sports and Physical Exercise Research Group (GIKAFIT), Department of Physical Education and Sport, Faculty of Education and Sport, University of the Basque Country (UPV/EHU), Vitoria-Gasteiz, Spain, 2 The Football Exchange, Research Institute for Sport and Exercise Sciences, Liverpool John Moores University, Liverpool, United Kingdom, 3 Department of Competitions and Mediacoach, LaLiga, Madrid, Spain

* ibai.errekagorri@ehu.eus

## Abstract

This study aimed to analyse the performance of 5,518 collective observations of the Spanish *LaLiga* teams for eight consecutive seasons (from 2011–12 to 2018–19), considering the final league ranking. The teams were divided into four groups: Europe (from 1st to 6th), Upper-Middle (from 7th to 11th), Lower-Middle (from 12th to 17th) and Relegation (from 18th to 20th). The variables recorded were: Passes, Successful Passes, Crosses, Shots, Goals, Corners, Fouls, Width, Length, Height, distance from the goalkeeper to the nearest defender (GkDef), total distance covered (TD) and number of points accumulated. The main results were that: 1) Europe, being superior to the rest of the groups, showed lower values of Length from 2015–16, and lower values of GkDef from 2014–15; 2) Upper-Middle showed lower values of Length from 2015–16; 3) Lower-Middle showed fewer Shots from 2013–14, and lower values of Length, GkDef and TD from 2014–15; and, 4) Relegation barely showed significant differences between seasons in any variable. The study concludes that the teams of the Europe, Upper-Middle and Relegation groups showed quite stable performance, while the teams of the Lower-Middle group presented a worsening in different dimensions as the seasons progressed. The information provided in this study makes it possible to have reference values that have characterized the performance of the teams for each group.

## Introduction

With the development of technology in sports and particularly in football, it has been possible to carry out more precise and objective studies about the performance of football players and teams during competition [1]. Nowadays, tracking systems (e.g., global navigation satellite systems or global positioning systems, local positioning systems, and semi-automatic video cameras) allow the analysis of kinematic variables (e.g., displacements, accelerations), as well as individual (e.g., heat maps) and collective (e.g., average positioning of the players) tactical

**Data Availability Statement:** All relevant data are within the manuscript.

**Funding:** The author(s) received no specific funding for this work.

**Competing interests:** The authors have declared that no competing interests exist.

variables of a team (e.g., distances between players and/or spaces covered by a group of players) based on the recorded positioning data [2–4]. The use of variables measuring physical and tactical aspects and covering individual player and teams' units is essential to evaluate the performance of players and teams in competition [5], and even to carry out longitudinal monitoring.

Previous studies explored the development of the game of football throughout the years [6–13]. Considering this longitudinal viewpoint, several studies have focused on analysing physical aspects [6, 7, 9, 10, 12]. In this regard, previous studies analysed the evolution of the English *Premier League* teams throughout seven seasons [7], considering the specific position of players [10] or the final ranking of teams at the end of the season [9]. Barnes et al. [7] reported that the distance covered by the teams in the English *Premier League* had not changed much throughout the seven years, this way increasing the number of high-intensity actions and accumulated distance, as well as the number of sprints and accumulated distance. Bradley et al. [9] showed that all the English *Premier League* teams increased the high-intensity distance covered when they were not in possession of the ball throughout the seven seasons. However, teams that finished fifth to eighth by the end of the season showed a slight increase in the short distance covered in high intensity when in possession of the ball compared to other teams. The teams ranked fifth to eighth also showed a significant increase in the distance covered while sprinting compared to other teams. Regarding the Spanish *LaLiga*, a recent study [12] showed a small decrease in the total distance covered by the teams throughout eight seasons. However, the Spanish *LaLiga* teams performed a higher number of high-intensity efforts as the seasons progressed, and the Upper-Middle ranked teams (from 6th to 10th) and Lower ranked teams (from 16th to 20th) covered a greater distance at high-intensity [12].

Nevertheless, the technical-tactical dimension has also received considerable attention in the scientific literature [7–13]. Thus, Barreira et al. [8] observed and recorded 45 matches and 6,791 attacks in the semi-finals and finals of the *UEFA Euro Championship* and the *FIFA World Cup* from 1982 to 2010. They concluded that similar attacks led by top-tier football teams had moved away from a more individualised behaviour, such as dribbling and feints in the centre of the pitch, to a more group-based performance, such as short passes and crosses into the box. Wallace & Norton [13] analysed the evolution of game-play in international competitions (*FIFA World Cups*) throughout a 44-year period. These researchers indicated that the speed of football had increased due to a significant boost in the number of passes in the last few years. As for domestic leagues, there has been an increase in the number of passes and their effectiveness in the English *Premier League* over seven seasons, mainly short and medium-distance passes [7]. During the seven-season period analysed in the study, the Tier A teams (from 1st to 4th) in the English *Premier League* demonstrated the greatest number of technical events and the highest levels of technical performance (i.e., number of passes and successful passes) [9]. However, the greatest increases in the technical parameters of passes made and received were shown by the Tier B teams (from 5th to 8th). On the other hand, a recent study [12] found that the Spanish *LaLiga* technical performance evolution throughout an eight-season period is dependent on the level of the teams. Top (from 1st to 5th), Upper-Middle (from 6th to 10th), and Lower-Middle (from 11th to 15th) ranked teams showed the greatest changes in different technical parameters as the seasons progressed (e.g., fewer shots, tackles or clearances, and more short passes, long passes, or aerial duels). On the contrary, Lower ranked (from 16th to 20th) teams showed more stable technical performance.

Nevertheless, it could be interesting to have more information about the evolution of the teams' performance in the Spanish men's top professional football division according to the final league ranking, especially the evolution of the teams' technical-tactical and physical performance [14]. Therefore, the present study aimed to analyse the Spanish *LaLiga* teams'

performance taking some key competitive performance variables into account over a continuous period of eight seasons according to the final league ranking.

## Materials and methods

### Sample

For the aim of this study, all teams' performances in the Spanish *LaLiga* across eight consecutive seasons (from 2011–12 to 2018–19) were analysed. All matches where the information required was not available were excluded, as well as matches where one or more players were sent off. As a result, out of a possible 6,080 performances (20 teams, each playing 38 matches throughout the eight seasons), a total of 5,518 performances were analysed, representing 90% of all the possible matches. During the eight-season period, 32 teams participated in the men's top professional football division from Spain. All the teams were divided into four groups according to the final league ranking each season: Europe (from 1st to 6th; n = 1,642), Upper-

**Table 1. Definitions of the variables for each dimension.**

| Dimensions | Variables | Definitions |
|---|---|---|
| Technical-Tactical | Passes | An intentional played ball from one player to another with any part of the body that is allowed in the rules of the game. When calculating this variable, the total number of successful and unsuccessful actions made by the team per match are considered. |
| | Successful Passes | A successful pass is one that reaches its recipient. To calculate this variable, the total number of successful exchanges of the ball between two players of the same team per match are considered. |
| | Crosses | Balls sent into the rival team's penalty box from a side area of the football pitch. When calculating this variable, the total number of successful and unsuccessful actions made by the team per match are considered. |
| | Shots | Attempt to score a goal, made with any part of the body that is allowed in the rules of the game. When calculating this variable, the total number of actions made by the team per match are considered. |
| Set Piece | Goals | Total number of points scored by each team per match. |
| | Corners | A kick that is performed on a set piece from the corner of the football pitch nearest to where the ball went out of the playing area. When calculating this variable, the total number of actions taken by the team per match are considered. |
| | Fouls | Any infringement that is penalised as foul play by the referee. When calculating this variable, the total number of actions received by the team per match are considered. |
| Collective Tactical Behaviour | Width | Mean team amplitude per match, considered as the distance (in m) between the two furthest-apart players of the same team along the amplitude of the pitch. To calculate this variable, the times in which the ball is out of play and the goalkeeper's activity are excluded. |
| | Length | Mean team depth per match, considered as the distance (in m) between the two furthest-apart players of the same team along the depth of the pitch. To calculate this variable, the times in which the ball is out of play and the goalkeeper's activity are excluded. |
| | Height | Mean team defence depth per match, considered as the distance (in m) between the furthest back player and the goal line he is defending. To calculate this variable, the times in which the ball is out of play and the goalkeeper's activity are excluded. |
| | GkDef | Mean distance (in m) from the goalkeeper to the nearest defender of the same team per match. To calculate this variable, the times in which the ball is out of play is excluded. |
| Physical | TD | Total distance covered (in m) by all the team's players that participated in the match, including the goalkeeper's activity. |

Middle (from 7th to 11th; n = 1,389), Lower-Middle (from 12th to 17th; n = 1,656) and Relegation (from 18th to 20th; n = 831). The data to carry out this study was collected in June 2019, after the end of the 2018–2019 season.

Data were obtained from the Spanish *Professional Football League*, which authorised the use of the variables included in this investigation. Following its ethical guidelines, this investigation does not include information that identifies football players. Data were treated in accordance with the Declaration of Helsinki, having been approved by the Ethics Committee on Humans (CEISH) of the *University of the Basque Country* (UPV/EHU).

### Variables

The variables used in this work were grouped into four dimensions: Technical-Tactical (Passes, Successful Passes, Crosses and Shots), Set Piece (Goals, Corners and Fouls), Collective Tactical Behaviour (Width, Length, Height and distance from the goalkeeper to the nearest defender (GkDef)) and Physical (total distance covered (TD)). Table 1 shows the definitions of these variables for each dimension. The number of points accumulated by the Spanish *LaLiga* teams was also calculated in each of the eight seasons.

### Procedures

Location and motion data were obtained using the computerised multi-camera tracking system *TRACAB* (*ChyronHego*, New York, USA), and events were obtained by the data company *OPTA* (*Opta Sports*, London, UK), both using *Mediacoach* software (*LaLiga*, Madrid, Spain). The reports were generated using *Mediacoach*, for the predefined performance indicators. The reliability of the *OPTA* system has been previously proved [15], and the reliability of the multi-camera tracking system *TRACAB* has also been tested for positioning and physical performance of the players [16]. The generated reports were exported into a *Microsoft Excel* spreadsheet (*Microsoft Corporation*, Washington, USA) to configure a matrix.

### Statistical analysis

The statistical analysis was conducted using the software *jamovi 2.4.8* [17] for *Windows*. A linear mixed model was carried out for each dependent variable in order to analyse the differences in teams' match performance according to the group and season. Group and season were considered as fixed effects and team as random effect. The Akaike information criterion (AIC) [18] and a likelihood ratio test [19] were used to select the model that best fitted each variable. The maximum likelihood (ML) estimation was used for model comparison and for the final model of each variable the best model again using restricted maximum likelihood (REML) estimation was refitted [19]. Marginal and conditional $R^2$ metrics [20] were provided for each linear mixed model as a measure of effect sizes. Marginal $R^2$ is concerned with variance explained by fixed effects, and conditional $R^2$ is concerned with variance explained by both fixed and random effects [20]. The level of significance was set at $p < 0.05$.

### Results

Table 2 shows the effects of season for each group and the effects of group on the variables of the Technical-Tactical dimension. In the Europe group, the teams showed fewer Crosses in 2017–18 (-6.309; p = 0.008) and 2018–19 (-4.559; p = 0.051) compared to the 2011–12 season. In the Upper-Middle, the teams showed fewer Crosses in 2018–19 (-4.835; p = 0.050) compared to the 2011–12 season. In the Lower-Middle, the teams showed fewer Crosses in 2016–17 (-3.563; p = 0.048) compared to the 2011–12 season, and fewer Shots in 2013–14 (-1.646;

**Table 2. Effects of season for each group and effects of group on the variables of the Technical-Tactical dimension.**

| | | Passes | | | Successful Passes | | | Crosses | | | Shots | | |
|---|---|---|---|---|---|---|---|---|---|---|---|---|---|
| Europe | **Fixed Effects** | **Estimate** | **SE** | **p** | **Estimate** | **SE** | **p** | **Estimate** | **SE** | **p** | **Estimate** | **SE** | **p** |
| | Intercept | 538.655 | 15.464 | <0.001 | 431.582 | 16.736 | <0.001 | 19.401 | 0.569 | <0.001 | 14.072 | 0.416 | <0.001 |
| | 2012–13–2011–12 | 0.272 | 61.799 | 0.997 | 3.004 | 66.889 | 0.964 | 0.016 | 2.267 | 0.994 | -0.406 | 1.657 | 0.808 |
| | 2013–14–2011–12 | -21.646 | 62.008 | 0.729 | -16.619 | 67.084 | 0.806 | -0.663 | 2.304 | 0.775 | -0.719 | 1.676 | 0.670 |
| | 2014–15–2011–12 | -25.669 | 61.811 | 0.680 | -21.100 | 66.901 | 0.754 | -0.061 | 2.270 | 0.979 | -1.206 | 1.659 | 0.471 |
| | 2015–16–2011–12 | 5.243 | 61.802 | 0.933 | 6.999 | 66.892 | 0.917 | -3.567 | 2.268 | 0.124 | -1.570 | 1.658 | 0.349 |
| | 2016–17–2011–12 | 14.796 | 61.796 | 0.812 | 24.839 | 66.886 | 0.712 | -4.202 | 2.267 | 0.071 | -0.822 | 1.657 | 0.623 |
| | 2017–18–2011–12 | 18.224 | 61.845 | 0.770 | 36.547 | 66.932 | 0.588 | -6.309 | 2.277 | 0.008 | -1.193 | 1.662 | 0.477 |
| | 2018–19–2011–12 | -19.907 | 61.808 | 0.749 | -10.282 | 66.898 | 0.879 | -4.559 | 2.269 | 0.051 | -1.745 | 1.658 | 0.299 |
| | **Random Effects** | **SD** | **Variance** | **ICC** | **SD** | **Variance** | **ICC** | **SD** | **Variance** | **ICC** | **SD** | **Variance** | **ICC** |
| | Team | 105.858 | 11,205.967 | 0.559 | 114.765 | 13,170.924 | 0.598 | 3.667 | 13.445 | 0.162 | 2.748 | 7.551 | 0.237 |
| | Residual | 94.116 | 8,857.737 | | 94.160 | 8,866.170 | | 8.348 | 69.687 | | 4.934 | 24.340 | |
| | Marginal R² / Conditional R² | 0.012 / 0.564 | | | 0.015 / 0.604 | | | 0.062 / 0.214 | | | 0.010 / 0.244 | | |
| Upper-Middle | **Fixed Effects** | **Estimate** | **SE** | **p** | **Estimate** | **SE** | **p** | **Estimate** | **SE** | **p** | **Estimate** | **SE** | **p** |
| | Intercept | 459.217 | 9.719 | <0.001 | 341.609 | 10.231 | <0.001 | 20.644 | 0.593 | <0.001 | 12.054 | 0.242 | <0.001 |
| | 2012–13–2011–12 | -21.570 | 38.886 | 0.583 | -9.211 | 40.930 | 0.823 | -1.710 | 2.375 | 0.477 | 0.790 | 0.970 | 0.422 |
| | 2013–14–2011–12 | -10.837 | 38.899 | 0.782 | 7.659 | 40.942 | 0.853 | -0.348 | 2.378 | 0.884 | 0.132 | 0.972 | 0.893 |
| | 2014–15–2011–12 | 15.168 | 38.895 | 0.699 | 21.233 | 40.938 | 0.608 | 0.617 | 2.377 | 0.797 | -0.383 | 0.971 | 0.696 |
| | 2015–16–2011–12 | 7.179 | 38.853 | 0.855 | 20.329 | 40.900 | 0.623 | -1.881 | 2.370 | 0.433 | -0.965 | 0.967 | 0.326 |
| | 2016–17–2011–12 | -15.707 | 38.864 | 0.689 | -4.711 | 40.911 | 0.909 | -1.118 | 2.372 | 0.641 | -0.872 | 0.968 | 0.374 |
| | 2017–18–2011–12 | -16.655 | 38.895 | 0.671 | -4.112 | 40.938 | 0.921 | -1.523 | 2.377 | 0.526 | -0.532 | 0.971 | 0.588 |
| | 2018–19–2011–12 | 15.618 | 38.869 | 0.691 | 41.143 | 40.916 | 0.322 | -4.835 | 2.373 | 0.050 | -0.771 | 0.968 | 0.432 |
| | **Random Effects** | **SD** | **Variance** | **ICC** | **SD** | **Variance** | **ICC** | **SD** | **Variance** | **ICC** | **SD** | **Variance** | **ICC** |
| | Team | 59.789 | 3,574.740 | 0.336 | 63.190 | 3,992.937 | 0.373 | 3.470 | 12.039 | 0.145 | 1.329 | 1.765 | 0.080 |
| | Residual | 83.994 | 7,054.980 | | 81.879 | 6,704.184 | | 8.428 | 71.024 | | 4.496 | 20.213 | |
| | Marginal R² / Conditional R² | 0.018 / 0.348 | | | 0.024 / 0.388 | | | 0.028 / 0.169 | | | 0.014 / 0.093 | | |
| Lower-Middle | **Fixed Effects** | **Estimate** | **SE** | **p** | **Estimate** | **SE** | **p** | **Estimate** | **SE** | **p** | **Estimate** | **SE** | **p** |
| | Intercept | 443.980 | 7.579 | <0.001 | 327.795 | 7.953 | <0.001 | 19.481 | 0.437 | <0.001 | 11.540 | 0.141 | <0.001 |
| | 2012–13–2011–12 | -6.686 | 30.305 | 0.827 | -8.816 | 31.799 | 0.783 | 0.791 | 1.745 | 0.653 | -1.035 | 0.564 | 0.074 |
| | 2013–14–2011–12 | -30.866 | 30.351 | 0.315 | -25.131 | 31.840 | 0.435 | 0.248 | 1.752 | 0.888 | -1.646 | 0.570 | 0.006 |
| | 2014–15–2011–12 | -36.742 | 30.305 | 0.233 | -30.793 | 31.799 | 0.339 | 0.669 | 1.745 | 0.704 | -2.044 | 0.564 | <0.001 |
| | 2015–16–2011–12 | -36.629 | 30.282 | 0.234 | -31.578 | 31.779 | 0.326 | -2.117 | 1.741 | 0.231 | -2.128 | 0.561 | <0.001 |
| | 2016–17–2011–12 | 20.206 | 30.304 | 0.509 | 34.296 | 31.798 | 0.287 | -3.563 | 1.745 | 0.048 | -1.432 | 0.564 | 0.015 |
| | 2017–18–2011–12 | -8.580 | 30.382 | 0.779 | 6.796 | 31.868 | 0.832 | -2.438 | 1.757 | 0.173 | -1.792 | 0.573 | 0.003 |
| | 2018–19–2011–12 | -20.031 | 30.287 | 0.512 | -7.966 | 31.783 | 0.803 | -0.583 | 1.742 | 0.739 | -1.038 | 0.561 | 0.072 |
| | **Random Effects** | **SD** | **Variance** | **ICC** | **SD** | **Variance** | **ICC** | **SD** | **Variance** | **ICC** | **SD** | **Variance** | **ICC** |
| | Team | 50.556 | 2,555.895 | 0.271 | 53.358 | 2,847.104 | 0.306 | 2.690 | 7.234 | 0.099 | 0.644 | 0.415 | 0.022 |
| | Residual | 83.011 | 6,890.763 | | 80.298 | 6,447.696 | | 8.107 | 65.729 | | 4.319 | 18.653 | |
| | Marginal R² / Conditional R² | 0.036 / 0.297 | | | 0.045 / 0.337 | | | 0.031 / 0.127 | | | 0.022 / 0.043 | | |

(*Continued*)

**Table 2.** (Continued)

| | Fixed Effects | Estimate | SE | p | Estimate | SE | p | Estimate | SE | p | Estimate | SE | p |
|---|---|---|---|---|---|---|---|---|---|---|---|---|---|
| Relegation | Intercept | 426.731 | 8.684 | <0.001 | 311.694 | 9.130 | <0.001 | 19.791 | 0.465 | <0.001 | 11.304 | 0.217 | <0.001 |
| | 2012–13–2011–12 | -7.955 | 34.753 | 0.822 | -3.391 | 36.533 | 0.927 | 3.855 | 1.864 | 0.055 | 1.393 | 0.869 | 0.129 |
| | 2013–14–2011–12 | 12.839 | 34.719 | 0.716 | 11.991 | 36.503 | 0.747 | 5.675 | 1.857 | 0.008 | -0.120 | 0.865 | 0.892 |
| | 2014–15–2011–12 | -33.577 | 34.733 | 0.348 | -32.347 | 36.516 | 0.389 | 2.297 | 1.860 | 0.235 | -1.752 | 0.867 | 0.061 |
| | 2015–16–2011–12 | 20.363 | 34.734 | 0.566 | 21.135 | 36.517 | 0.571 | 3.759 | 1.860 | 0.060 | 0.766 | 0.867 | 0.390 |
| | 2016–17–2011–12 | -23.200 | 34.715 | 0.514 | -17.768 | 36.500 | 0.633 | -0.758 | 1.857 | 0.689 | -1.391 | 0.865 | 0.128 |
| | 2017–18–2011–12 | 39.490 | 34.744 | 0.272 | 51.978 | 36.525 | 0.174 | 0.643 | 1.862 | 0.734 | -0.022 | 0.868 | 0.980 |
| | 2018–19–2011–12 | -1.156 | 34.753 | 0.974 | 13.005 | 36.533 | 0.727 | 2.685 | 1.864 | 0.169 | 0.143 | 0.869 | 0.872 |
| | **Random Effects** | **SD** | **Variance** | **ICC** | **SD** | **Variance** | **ICC** | **SD** | **Variance** | **ICC** | **SD** | **Variance** | **ICC** |
| | Team | 40.113 | 1,609.042 | 0.188 | 42.584 | 1,813.360 | 0.219 | 1.789 | 3.201 | 0.044 | 0.749 | 0.561 | 0.028 |
| | Residual | 83.324 | 6,942.861 | | 80.406 | 6,465.090 | | 8.300 | 68.889 | | 4.428 | 19.609 | |
| | Marginal R² / Conditional R² | 0.054 / 0.232 | | | 0.065 / 0.270 | | | 0.055 / 0.097 | | | 0.044 / 0.071 | | |
| All seasons | **Fixed Effects** | **Estimate** | **SE** | **p** | **Estimate** | **SE** | **p** | **Estimate** | **SE** | **p** | **Estimate** | **SE** | **p** |
| | Intercept | 467.184 | 5.770 | <0.001 | 353.206 | 6.201 | <0.001 | 19.826 | 0.302 | <0.001 | 12.245 | 0.151 | <0.001 |
| | Upper-Middle—Europe | -79.536 | 15.016 | <0.001 | -90.069 | 16.137 | <0.001 | 1.250 | 0.785 | 0.114 | -2.030 | 0.394 | <0.001 |
| | Lower-Middle—Europe | -94.736 | 14.320 | <0.001 | -103.856 | 15.389 | <0.001 | 0.089 | 0.749 | 0.905 | -2.541 | 0.376 | <0.001 |
| | Relegation—Europe | -112.009 | 17.532 | <0.001 | -119.975 | 18.842 | <0.001 | 0.400 | 0.917 | 0.663 | -2.779 | 0.460 | <0.001 |
| | **Random Effects** | **SD** | **Variance** | **ICC** | **SD** | **Variance** | **ICC** | **SD** | **Variance** | **ICC** | **SD** | **Variance** | **ICC** |
| | Team | 68.529 | 4,696.183 | 0.384 | 73.938 | 5,466.775 | 0.430 | 3.380 | 11.424 | 0.143 | 1.666 | 2.777 | 0.117 |
| | Residual | 86.747 | 7,524.955 | | 85.055 | 7,234.320 | | 8.289 | 68.715 | | 4.570 | 20.883 | |
| | Marginal R² / Conditional R² | 0.135 / 0.467 | | | 0.152 / 0.517 | | | 0.003 / 0.145 | | | 0.051 / 0.162 | | |

Note: SE is Standard Error; SD is Standard Deviation; ICC is Intraclass Correlation Coefficient. Statistical significance set at p<0.05.

**Table 3. Effects of season for each group and effects of group on the variables of the Set Piece dimension.**

| | | Goals | | | Corners | | | Fouls | | |
|---|---|---|---|---|---|---|---|---|---|---|
| Europe | Fixed Effects | Estimate | SE | p | Estimate | SE | p | Estimate | SE | p |
| | Intercept | 2.008 | 0.097 | <0.001 | 5.691 | 0.132 | <0.001 | 14.326 | 0.174 | <0.001 |
| | 2012–13–2011–12 | 0.007 | 0.385 | 0.985 | -0.426 | 0.524 | 0.421 | -0.280 | 0.688 | 0.687 |
| | 2013–14–2011–12 | 0.063 | 0.391 | 0.872 | 0.152 | 0.539 | 0.780 | -0.688 | 0.711 | 0.339 |
| | 2014–15–2011–12 | 0.021 | 0.385 | 0.957 | -0.685 | 0.525 | 0.200 | -0.355 | 0.690 | 0.609 |
| | 2015–16–2011–12 | -0.122 | 0.385 | 0.752 | -0.650 | 0.524 | 0.223 | -1.585 | 0.689 | 0.027 |
| | 2016–17–2011–12 | 0.041 | 0.385 | 0.915 | -1.338 | 0.523 | 0.015 | -0.478 | 0.687 | 0.491 |
| | 2017–18–2011–12 | -0.043 | 0.387 | 0.913 | -1.039 | 0.529 | 0.057 | -1.390 | 0.696 | 0.052 |
| | 2018–19–2011–12 | -0.427 | 0.385 | 0.274 | -0.876 | 0.525 | 0.103 | -1.189 | 0.690 | 0.092 |
| | **Random Effects** | **SD** | **Variance** | **ICC** | **SD** | **Variance** | **ICC** | **SD** | **Variance** | **ICC** |
| | Team | 0.618 | 0.382 | 0.148 | 0.757 | 0.573 | 0.061 | 0.960 | 0.921 | 0.050 |
| | Residual | 1.481 | 2.193 | | 2.971 | 8.829 | | 4.197 | 17.616 | |
| | Marginal R² / Conditional R² | 0.009 / 0.156 | | | 0.022 / 0.082 | | | 0.016 / 0.065 | | |

(*Continued*)

**Table 3.** (Continued)

| Upper-Middle | Fixed Effects | Estimate | SE | p | Estimate | SE | p | Estimate | SE | p |
|---|---|---|---|---|---|---|---|---|---|---|
| | Intercept | 1.190 | 0.032 | <0.001 | 5.146 | 0.141 | <0.001 | 14.132 | 0.183 | <0.001 |
| | 2012–13–2011–12 | 0.314 | 0.128 | 0.020 | -0.012 | 0.566 | 0.984 | 0.118 | 0.732 | 0.873 |
| | 2013–14–2011–12 | 0.189 | 0.129 | 0.152 | -0.279 | 0.567 | 0.626 | 0.100 | 0.734 | 0.892 |
| | 2014–15–2011–12 | 0.143 | 0.129 | 0.276 | -0.007 | 0.567 | 0.990 | 0.412 | 0.733 | 0.578 |
| | 2015–16–2011–12 | 0.097 | 0.126 | 0.448 | -0.440 | 0.564 | 0.441 | 0.676 | 0.728 | 0.360 |
| | 2016–17–2011–12 | 0.312 | 0.127 | 0.020 | -0.754 | 0.564 | 0.191 | -0.821 | 0.729 | 0.269 |
| | 2017–18–2011–12 | 0.118 | 0.129 | 0.365 | -0.327 | 0.567 | 0.568 | -0.330 | 0.733 | 0.656 |
| | 2018–19–2011–12 | 0.157 | 0.127 | 0.226 | -1.113 | 0.565 | 0.057 | 0.402 | 0.730 | 0.586 |
| | **Random Effects** | **SD** | **Variance** | **ICC** | **SD** | **Variance** | **ICC** | **SD** | **Variance** | **ICC** |
| | Team | 0.070 | 0.005 | 0.004 | 0.766 | 0.587 | 0.074 | 0.915 | 0.838 | 0.046 |
| | Residual | 1.115 | 1.243 | | 2.711 | 7.350 | | 4.151 | 17.228 | |
| | Marginal $R^2$ / Conditional $R^2$ | 0.008 / 0.012 | | | 0.017 / 0.090 | | | 0.011 / 0.057 | | |
| Lower-Middle | **Fixed Effects** | **Estimate** | **SE** | **p** | **Estimate** | **SE** | **p** | **Estimate** | **SE** | **p** |
| | Intercept | 1.131 | 0.028 | <0.001 | 4.847 | 0.090 | <0.001 | 13.862 | 0.190 | <0.001 |
| | 2012–13–2011–12 | -0.221 | 0.113 | 0.059 | 0.107 | 0.358 | 0.766 | -0.925 | 0.758 | 0.229 |
| | 2013–14–2011–12 | -0.323 | 0.115 | 0.008 | -0.000 | 0.361 | 0.999 | -0.527 | 0.763 | 0.494 |
| | 2014–15–2011–12 | -0.368 | 0.113 | 0.002 | -0.365 | 0.358 | 0.314 | -0.696 | 0.758 | 0.364 |
| | 2015–16–2011–12 | -0.117 | 0.112 | 0.302 | -0.620 | 0.356 | 0.089 | -1.258 | 0.756 | 0.104 |
| | 2016–17–2011–12 | -0.034 | 0.113 | 0.766 | -0.734 | 0.358 | 0.047 | 0.577 | 0.758 | 0.451 |
| | 2017–18–2011–12 | -0.092 | 0.116 | 0.434 | -0.595 | 0.363 | 0.109 | -0.633 | 0.765 | 0.412 |
| | 2018–19–2011–12 | -0.093 | 0.112 | 0.415 | -0.460 | 0.356 | 0.204 | -1.887 | 0.757 | 0.017 |
| | **Random Effects** | **SD** | **Variance** | **ICC** | **SD** | **Variance** | **ICC** | **SD** | **Variance** | **ICC** |
| | Team | 0.062 | 0.004 | 0.003 | 0.429 | 0.184 | 0.026 | 1.123 | 1.261 | 0.073 |
| | Residual | 1.096 | 1.200 | | 2.631 | 6.921 | | 4.007 | 16.058 | |
| | Marginal $R^2$ / Conditional $R^2$ | 0.013 / 0.016 | | | 0.013 / 0.038 | | | 0.028 / 0.099 | | |
| Relegation | **Fixed Effects** | **Estimate** | **SE** | **p** | **Estimate** | **SE** | **p** | **Estimate** | **SE** | **p** |
| | Intercept | 0.973 | 0.036 | <0.001 | 4.822 | 0.137 | <0.001 | 13.518 | 0.329 | <0.001 |
| | 2012–13–2011–12 | 0.187 | 0.144 | 0.213 | 0.744 | 0.549 | 0.194 | -1.099 | 1.318 | 0.416 |
| | 2013–14–2011–12 | -0.007 | 0.143 | 0.961 | 1.444 | 0.547 | 0.018 | -2.555 | 1.316 | 0.070 |
| | 2014–15–2011–12 | -0.115 | 0.143 | 0.436 | -0.338 | 0.548 | 0.546 | -2.003 | 1.317 | 0.148 |
| | 2015–16–2011–12 | 0.175 | 0.143 | 0.240 | 0.440 | 0.548 | 0.434 | -2.240 | 1.317 | 0.108 |
| | 2016–17–2011–12 | 0.021 | 0.143 | 0.886 | -0.528 | 0.547 | 0.349 | -3.033 | 1.315 | 0.035 |
| | 2017–18–2011–12 | -0.146 | 0.144 | 0.325 | 0.105 | 0.549 | 0.851 | -1.126 | 1.317 | 0.405 |
| | 2018–19–2011–12 | 0.137 | 0.144 | 0.356 | -0.286 | 0.549 | 0.609 | -2.237 | 1.318 | 0.109 |
| | **Random Effects** | **SD** | **Variance** | **ICC** | **SD** | **Variance** | **ICC** | **SD** | **Variance** | **ICC** |
| | Team | 0.069 | 0.005 | 0.005 | 0.512 | 0.262 | 0.039 | 1.447 | 2.094 | 0.107 |
| | Residual | 0.950 | 0.902 | | 2.552 | 6.512 | | 4.186 | 17.526 | |
| | Marginal $R^2$ / Conditional $R^2$ | 0.015 / 0.020 | | | 0.053 / 0.090 | | | 0.041 / 0.144 | | |
| All seasons | **Fixed Effects** | **Estimate** | **SE** | **p** | **Estimate** | **SE** | **p** | **Estimate** | **SE** | **p** |
| | Intercept | 1.325 | 0.032 | <0.001 | 5.123 | 0.070 | <0.001 | 13.959 | 0.110 | <0.001 |
| | Upper-Middle—Europe | -0.820 | 0.084 | <0.001 | -0.538 | 0.181 | 0.004 | -0.196 | 0.285 | 0.494 |
| | Lower-Middle—Europe | -0.877 | 0.080 | <0.001 | -0.837 | 0.173 | <0.001 | -0.469 | 0.272 | 0.087 |
| | Relegation—Europe | -1.037 | 0.098 | <0.001 | -0.859 | 0.212 | <0.001 | -0.813 | 0.333 | 0.016 |
| | **Random Effects** | **SD** | **Variance** | **ICC** | **SD** | **Variance** | **ICC** | **SD** | **Variance** | **ICC** |
| | Team | 0.331 | 0.109 | 0.070 | 0.705 | 0.497 | 0.062 | 1.131 | 1.278 | 0.070 |
| | Residual | 1.209 | 1.462 | | 2.745 | 7.534 | | 4.128 | 17.037 | |
| | Marginal $R^2$ / Conditional $R^2$ | 0.098 / 0.161 | | | 0.016 / 0.077 | | | 0.004 / 0.074 | | |

Note: SE is Standard Error; SD is Standard Deviation; ICC is Intraclass Correlation Coefficient. Statistical significance set at $p < 0.05$.

p = 0.006), 2014–15 (-2.044; p<0.001), 2015–16 (-2.128; p<0.001), 2016–17 (-1.432; p = 0.015) and 2017–18 (-1.792; p = 0.003) compared to the 2011–12 season. In the Relegation group, the teams showed more Crosses in 2013–14 (5.675; p = 0.008) compared to the 2011–12 season. Likewise, Europe showed more Passes than Upper-Middle (79.536; p<0.001), Lower-Middle (94.736; p<0.001) and Relegation (112.009; p<0.001), more Successful Passes than Upper-Middle (90.069; p<0.001), Lower-Middle (103.856; p<0.001) and Relegation (119.975; p<0.001), and more Shots than Upper-Middle (2.030; p<0.001), Lower-Middle (2.541; p<0.001) and Relegation (2.779; p<0.001) during the whole period analysed.

Table 3 shows the effects of season for each group and the effects of group on the variables of the Set Piece dimension. In the Europe group, the teams showed fewer Corners in 2016–17 (-1.338; p = 0.015) compared to the 2011–12 season, and fewer Fouls in 2015–16 (-1.585; p = 0.027) and 2017–18 (-1.390; p = 0.052) compared to the 2011–12 season. In the Upper-Middle, the teams showed more Goals in 2012–13 (0.314; p = 0.020) and 2016–17 (0.312; p = 0.020) compared to the 2011–12 season. In the Lower-Middle, the teams showed fewer Goals in 2013–14 (-0.323; p = 0.008) and 2014–15 (-0.368; p = 0.002) compared to the 2011–12 season, fewer Corners in 2016–17 (-0.734; p = 0.047) compared to the 2011–12 season, and fewer Fouls in 2018–19 (-1.887; p = 0.017) compared to the 2011–12 season. In the Relegation group, the teams showed more Corners in 2013–14 (1.444; p = 0.018) compared to the 2011–12 season, and fewer Fouls in 2016–17 (-3.033; p = 0.035) compared to the 2011–12 season. Likewise, Europe showed more Goals than Upper-Middle (0.820; p<0.001), Lower-Middle (0.877; p<0.001) and Relegation (1.037; p<0.001), more Corners than Upper-Middle (0.538; p = 0.004), Lower-Middle (0.837; p<0.001) and Relegation (0.859; p<0.001), and more Fouls than Relegation (0.813; p = 0.016) during the whole period analysed.

Table 4 shows the effects of season for each group and the effects of group on the variables of the Collective Tactical Behaviour dimension. In the Europe group, the teams showed lower values of Length in 2015–16 (-1.665; p = 0.015), 2016–17 (-1.613; p = 0.019), 2017–18 (-1.930; p = 0.006) and 2018–19 (-2.276; p = 0.001) compared to the season 2011–12, and lower values of GkDef in 2014–15 (-3.190; p = 0.001), 2015–16 (-3.169; p = 0.001), 2016–17 (-2.722; p = 0.005), 2017–18 (-2.633; p = 0.007) and 2018–19 (-2.487; p = 0.010) compared to the season 2011–12. In the Upper-Middle group, the teams showed lower values of Length in 2015–16 (-1.622; p = 0.001), 2016–17 (-2.706; p<0.001), 2017–18 (-2.463; p<0.001) and 2018–19 (-1.952; p<0.001) compared to the season 2011–12. In the Lower-Middle group, the teams showed lower values of Length in 2014–15 (-1.218; p = 0.040), 2015–16 (-1.660; p = 0.006), 2016–17 (-1.609; p = 0.008), 2017–18 (-2.211; p<0.001) and 2018–19 (-2.542; p<0.001) compared to the season 2011–12, lower values of Height in 2014–15 (-1.407; p = 0.040) compared to the season 2011–12, and lower values of GkDef in 2014–15 (-2.002; p<0.001), 2015–16 (-1.668; p = 0.002), 2016–17 (-1.839; p<0.001), 2017–18 (-1.747; p = 0.001) and 2018–19 (-1.371; p = 0.009) compared to the season 2011–12. In the Relegation group, the teams showed lower values of Length in 2016–17 (-1.851; p = 0.006) and 2018–19 (-1.263; p = 0.044) compared to the 2011–12 season, lower values of Height in 2014–15 (-1.893; p = 0.043) compared to the 2011–12 season, and lower values of GkDef in 2014–15 (-3.638; p<0.001) and 2015–16 (-2.506; p = 0.009) compared to the 2011–12 season. Likewise, Europe showed higher values of Width than Upper-Middle (0.928; p = 0.009), Lower-Middle (1.010; p = 0.003) and Relegation (1.373; p = 0.001), higher values of Length than Upper-Middle (0.667; p = 0.010), Lower-Middle (0.756; p = 0.002) and Relegation (1.055; p<0.001), higher values of Height than Upper-Middle (1.164; p<0.001), Lower-Middle (1.412; p<0.001) and Relegation (1.726; p<0.001), and higher values of GkDef than Upper-Middle (1.175; p<0.001), Lower-Middle (0.985; p = 0.002) and Relegation (0.871; p = 0.026) during the whole period analysed.

**Table 4. Effects of season for each group and effects of group on the variables of the Collective Tactical Behaviour dimension.**

| | | Width | | | Length | | | Height | | | GkDef | | |
|---|---|---|---|---|---|---|---|---|---|---|---|---|---|
| Europe | **Fixed Effects** | **Estimate** | **SE** | **p** | **Estimate** | **SE** | **p** | **Estimate** | **SE** | **p** | **Estimate** | **SE** | **p** |
| | Intercept | 44.250 | 0.321 | <0.001 | 37.475 | 0.165 | <0.001 | 38.402 | 0.313 | <0.001 | 25.730 | 0.230 | <0.001 |
| | 2012–13–2011–12 | 0.335 | 1.283 | 0.795 | -0.285 | 0.657 | 0.667 | -0.311 | 1.251 | 0.805 | 0.326 | 0.919 | 0.724 |
| | 2013–14–2011–12 | -0.219 | 1.288 | 0.866 | -0.716 | 0.662 | 0.286 | -0.509 | 1.262 | 0.688 | -0.718 | 0.924 | 0.442 |
| | 2014–15–2011–12 | -0.679 | 1.284 | 0.600 | -1.056 | 0.658 | 0.116 | -2.192 | 1.251 | 0.087 | -3.190 | 0.919 | 0.001 |
| | 2015–16–2011–12 | -0.218 | 1.283 | 0.866 | -1.665 | 0.657 | 0.015 | -1.411 | 1.251 | 0.266 | -3.169 | 0.919 | 0.001 |
| | 2016–17–2011–12 | 0.892 | 1.283 | 0.491 | -1.613 | 0.657 | 0.019 | -1.277 | 1.250 | 0.313 | -2.722 | 0.919 | 0.005 |
| | 2017–18–2011–12 | 0.509 | 1.284 | 0.694 | -1.930 | 0.658 | 0.006 | -2.043 | 1.253 | 0.111 | -2.633 | 0.920 | 0.007 |
| | 2018–19–2011–12 | -0.599 | 1.284 | 0.643 | -2.276 | 0.658 | 0.001 | -1.369 | 1.251 | 0.280 | -2.487 | 0.919 | 0.010 |
| | **Random Effects** | **SD** | **Variance** | **ICC** | **SD** | **Variance** | **ICC** | **SD** | **Variance** | **ICC** | **SD** | **Variance** | **ICC** |
| | Team | 2.199 | 4.834 | 0.561 | 1.110 | 1.232 | 0.354 | 2.098 | 4.401 | 0.301 | 1.560 | 2.433 | 0.411 |
| | Residual | 1.944 | 3.780 | | 1.499 | 2.246 | | 3.199 | 10.233 | | 1.866 | 3.483 | |
| | Marginal $R^2$ / Conditional $R^2$ | 0.030 / 0.574 | | | 0.143 / 0.447 | | | 0.036 / 0.326 | | | 0.241 / 0.553 | | |
| Upper-Middle | **Fixed Effects** | **Estimate** | **SE** | **p** | **Estimate** | **SE** | **p** | **Estimate** | **SE** | **p** | **Estimate** | **SE** | **p** |
| | Intercept | 43.324 | 0.263 | <0.001 | 36.807 | 0.116 | <0.001 | 37.234 | 0.266 | <0.001 | 24.552 | 0.201 | <0.001 |
| | 2012–13–2011–12 | 0.376 | 1.051 | 0.723 | -0.196 | 0.463 | 0.675 | -0.122 | 1.065 | 0.909 | 0.120 | 0.802 | 0.882 |
| | 2013–14–2011–12 | 0.729 | 1.051 | 0.493 | -0.737 | 0.463 | 0.121 | 0.045 | 1.066 | 0.966 | 0.692 | 0.802 | 0.395 |
| | 2014–15–2011–12 | -0.086 | 1.051 | 0.935 | -0.786 | 0.463 | 0.099 | -0.110 | 1.066 | 0.918 | -0.944 | 0.802 | 0.248 |
| | 2015–16–2011–12 | 0.171 | 1.050 | 0.871 | -1.622 | 0.462 | 0.001 | -0.170 | 1.064 | 0.874 | -0.881 | 0.802 | 0.280 |
| | 2016–17–2011–12 | -0.518 | 1.050 | 0.625 | -2.706 | 0.462 | <0.001 | 0.352 | 1.065 | 0.743 | -0.691 | 0.802 | 0.395 |
| | 2017–18–2011–12 | -0.075 | 1.051 | 0.943 | -2.463 | 0.463 | <0.001 | 0.392 | 1.066 | 0.716 | -0.240 | 0.802 | 0.767 |
| | 2018–19–2011–12 | 0.479 | 1.050 | 0.651 | -1.952 | 0.462 | <0.001 | -0.066 | 1.065 | 0.951 | -1.341 | 0.802 | 0.104 |
| | **Random Effects** | **SD** | **Variance** | **ICC** | **SD** | **Variance** | **ICC** | **SD** | **Variance** | **ICC** | **SD** | **Variance** | **ICC** |
| | Team | 1.623 | 2.633 | 0.374 | 0.674 | 0.454 | 0.141 | 1.611 | 2.594 | 0.236 | 1.237 | 1.531 | 0.364 |
| | Residual | 2.099 | 4.404 | | 1.665 | 2.771 | | 2.896 | 8.386 | | 1.636 | 2.677 | |
| | Marginal $R^2$ / Conditional $R^2$ | 0.019 / 0.386 | | | 0.222 / 0.331 | | | 0.004 / 0.239 | | | 0.085 / 0.418 | | |
| Lower-Middle | **Fixed Effects** | **Estimate** | **SE** | **p** | **Estimate** | **SE** | **p** | **Estimate** | **SE** | **p** | **Estimate** | **SE** | **p** |
| | Intercept | 43.241 | 0.206 | <0.001 | 36.718 | 0.144 | <0.001 | 36.987 | 0.166 | <0.001 | 24.744 | 0.126 | <0.001 |
| | 2012–13–2011–12 | -1.264 | 0.825 | 0.133 | -0.914 | 0.574 | 0.119 | 0.048 | 0.661 | 0.943 | -0.564 | 0.502 | 0.268 |
| | 2013–14–2011–12 | -0.362 | 0.826 | 0.664 | -0.855 | 0.575 | 0.145 | -0.085 | 0.664 | 0.898 | -0.105 | 0.503 | 0.836 |
| | 2014–15–2011–12 | -1.187 | 0.825 | 0.158 | -1.218 | 0.574 | 0.040 | -1.407 | 0.661 | 0.040 | -2.002 | 0.502 | <0.001 |
| | 2015–16–2011–12 | -0.621 | 0.824 | 0.456 | -1.660 | 0.573 | 0.006 | -0.262 | 0.660 | 0.694 | -1.668 | 0.502 | 0.002 |
| | 2016–17–2011–12 | 0.413 | 0.825 | 0.619 | -1.609 | 0.574 | 0.008 | -0.211 | 0.661 | 0.751 | -1.839 | 0.502 | <0.001 |
| | 2017–18–2011–12 | -0.468 | 0.827 | 0.574 | -2.211 | 0.576 | <0.001 | -0.408 | 0.666 | 0.543 | -1.747 | 0.504 | 0.001 |
| | 2018–19–2011–12 | -0.951 | 0.824 | 0.256 | -2.542 | 0.573 | <0.001 | -0.056 | 0.660 | 0.932 | -1.371 | 0.502 | 0.009 |
| | **Random Effects** | **SD** | **Variance** | **ICC** | **SD** | **Variance** | **ICC** | **SD** | **Variance** | **ICC** | **SD** | **Variance** | **ICC** |
| | Team | 1.382 | 1.911 | 0.298 | 0.949 | 0.900 | 0.229 | 1.036 | 1.073 | 0.115 | 0.824 | 0.680 | 0.203 |
| | Residual | 2.123 | 4.507 | | 1.741 | 3.032 | | 2.879 | 8.291 | | 1.632 | 2.664 | |
| | Marginal $R^2$ / Conditional $R^2$ | 0.044 / 0.329 | | | 0.127 / 0.327 | | | 0.021 / 0.133 | | | 0.146 / 0.319 | | |

(*Continued*)

**Table 4.** (Continued)

| | Fixed Effects | Estimate | SE | p | Estimate | SE | p | Estimate | SE | p | Estimate | SE | p |
|---|---|---|---|---|---|---|---|---|---|---|---|---|---|
| Relegation | Intercept | 42.877 | 0.212 | <0.001 | 36.419 | 0.145 | <0.001 | 36.672 | 0.216 | <0.001 | 24.856 | 0.212 | <0.001 |
| | 2012–13–2011–12 | -0.574 | 0.848 | 0.508 | -0.042 | 0.579 | 0.943 | 1.395 | 0.863 | 0.125 | 0.737 | 0.847 | 0.397 |
| | 2013–14–2011–12 | 0.127 | 0.848 | 0.883 | 0.159 | 0.578 | 0.787 | 0.427 | 0.861 | 0.627 | -0.436 | 0.847 | 0.613 |
| | 2014–15–2011–12 | -0.544 | 0.848 | 0.530 | -0.114 | 0.579 | 0.846 | -1.893 | 0.862 | 0.043 | -3.638 | 0.847 | <0.001 |
| | 2015–16–2011–12 | -0.077 | 0.848 | 0.928 | -0.943 | 0.579 | 0.123 | 0.105 | 0.862 | 0.904 | -2.506 | 0.847 | 0.009 |
| | 2016–17–2011–12 | -0.284 | 0.848 | 0.742 | -1.851 | 0.578 | 0.006 | 0.469 | 0.861 | 0.593 | -1.662 | 0.846 | 0.067 |
| | 2017–18–2011–12 | 0.648 | 0.848 | 0.456 | -0.636 | 0.579 | 0.288 | 0.461 | 0.863 | 0.600 | -1.266 | 0.847 | 0.154 |
| | 2018–19–2011–12 | 0.707 | 0.848 | 0.417 | -1.263 | 0.579 | 0.044 | -0.519 | 0.863 | 0.556 | -1.569 | 0.847 | 0.083 |
| | **Random Effects** | **SD** | **Variance** | **ICC** | **SD** | **Variance** | **ICC** | **SD** | **Variance** | **ICC** | **SD** | **Variance** | **ICC** |
| | Team | 0.979 | 0.958 | 0.187 | 0.641 | 0.411 | 0.115 | 0.939 | 0.881 | 0.098 | 0.996 | 0.992 | 0.255 |
| | Residual | 2.041 | 4.167 | | 1.780 | 3.169 | | 2.846 | 8.099 | | 1.702 | 2.895 | |
| | Marginal R² / Conditional R² | 0.038 / 0.218 | | | 0.113 / 0.215 | | | 0.082 / 0.172 | | | 0.306 / 0.483 | | |
| All seasons | **Fixed Effects** | **Estimate** | **SE** | **p** | **Estimate** | **SE** | **p** | **Estimate** | **SE** | **p** | **Estimate** | **SE** | **p** |
| | Intercept | 43.423 | 0.135 | <0.001 | 36.854 | 0.098 | <0.001 | 37.323 | 0.133 | <0.001 | 24.969 | 0.127 | <0.001 |
| | Upper-Middle—Europe | -0.928 | 0.352 | 0.009 | -0.667 | 0.254 | 0.010 | -1.164 | 0.346 | <0.001 | -1.175 | 0.331 | <0.001 |
| | Lower-Middle—Europe | -1.010 | 0.336 | 0.003 | -0.756 | 0.242 | 0.002 | -1.412 | 0.330 | <0.001 | -0.985 | 0.316 | 0.002 |
| | Relegation—Europe | -1.373 | 0.411 | 0.001 | -1.055 | 0.297 | <0.001 | -1.726 | 0.404 | <0.001 | -0.871 | 0.387 | 0.026 |
| | **Random Effects** | **SD** | **Variance** | **ICC** | **SD** | **Variance** | **ICC** | **SD** | **Variance** | **ICC** | **SD** | **Variance** | **ICC** |
| | Team | 1.607 | 2.582 | 0.380 | 1.153 | 1.329 | 0.326 | 1.531 | 2.345 | 0.209 | 1.520 | 2.309 | 0.439 |
| | Residual | 2.053 | 4.214 | | 1.659 | 2.753 | | 2.977 | 8.864 | | 1.716 | 2.946 | |
| | Marginal R² / Conditional R² | 0.036 / 0.402 | | | 0.034 / 0.349 | | | 0.037 / 0.239 | | | 0.042 / 0.463 | | |

Note: SE is Standard Error; SD is Standard Deviation; ICC is Intraclass Correlation Coefficient. Statistical significance set at p<0.05.

Table 5 shows the effects of season for each group and the effects of group on the variable of the Physical dimension. In the Europe group, the teams showed lower values of TD in 2018–19 (-3,200.245; p = 0.050) compared to the 2011–12 season. In the Lower-Middle group, the teams showed lower values of TD in 2014–15 (-3,741.391; p = 0.011), 2015–16 (-3,278.483;

**Table 5.  Effects of season for each group and effects of group on the variable of the Physical dimension.**

| | | | TD | | |
|---|---|---|---|---|---|
| Europe | **Fixed Effects** | | **Estimate** | **SE** | **p** |
| | Intercept | | 109,316.536 | 395.921 | <0.001 |
| | 2012–13–2011–12 | | -664.549 | 1,580.860 | 0.676 |
| | 2013–14–2011–12 | | -240.763 | 1,591.088 | 0.880 |
| | 2014–15–2011–12 | | -2,306.592 | 1,581.493 | 0.153 |
| | 2015–16–2011–12 | | -1,503.857 | 1,581.014 | 0.347 |
| | 2016–17–2011–12 | | -1,196.475 | 1,580.702 | 0.454 |
| | 2017–18–2011–12 | | -1,728.212 | 1,583.236 | 0.282 |
| | 2018–19–2011–12 | | -3,200.245 | 1,581.420 | 0.050 |
| | **Random Effects** | | **SD** | **Variance** | **ICC** |
| | Team | | 2,677.165 | 7,167,211.081 | 0.381 |
| | Residual | | 3,411.303 | 11,636,987.536 | |
| | Marginal R² / Conditional R² | | 0.050 / 0.412 | | |

*(Continued)*

**Table 5.** (Continued)

| Upper-Middle | Fixed Effects | Estimate | SE | p |
|---|---|---|---|---|
| | Intercept | 110,839.230 | 417.130 | <0.001 |
| | 2012–13–2011–12 | -899.901 | 1,668.878 | 0.593 |
| | 2013–14–2011–12 | -952.604 | 1,669.421 | 0.572 |
| | 2014–15–2011–12 | -2,835.551 | 1,669.253 | 0.099 |
| | 2015–16–2011–12 | -2,892.991 | 1,667.468 | 0.092 |
| | 2016–17–2011–12 | -2,118.098 | 1,667.958 | 0.213 |
| | 2017–18–2011–12 | -2,984.391 | 1,669.250 | 0.083 |
| | 2018–19–2011–12 | -3,191.599 | 1,668.314 | 0.065 |
| | **Random Effects** | **SD** | **Variance** | **ICC** |
| | Team | 2,566.647 | 6,587,675.591 | 0.338 |
| | Residual | 3,588.856 | 12,879,889.799 | |
| | Marginal $R^2$ / Conditional $R^2$ | | 0.016 / 0.379 | |
| Lower-Middle | **Fixed Effects** | **Estimate** | **SE** | **p** |
| | Intercept | 109,912.326 | 349.425 | <0.001 |
| | 2012–13–2011–12 | 22.669 | 1,397.123 | 0.987 |
| | 2013–14–2011–12 | -1,846.974 | 1,399.127 | 0.194 |
| | 2014–15–2011–12 | -3,741.391 | 1,397.098 | 0.011 |
| | 2015–16–2011–12 | -3,278.483 | 1,396.099 | 0.024 |
| | 2016–17–2011–12 | -3,793.554 | 1,397.074 | 0.010 |
| | 2017–18–2011–12 | -2,090.699 | 1,400.508 | 0.143 |
| | 2018–19–2011–12 | -2,863.177 | 1,396.528 | 0.047 |
| | **Random Effects** | **SD** | **Variance** | **ICC** |
| | Team | 2,335.007 | 5,452,259.170 | 0.281 |
| | Residual | 3,735.540 | 13,954,255.658 | |
| | Marginal $R^2$ / Conditional $R^2$ | | 0.097 / 0.351 | |
| Relegation | **Fixed Effects** | **Estimate** | **SE** | **p** |
| | Intercept | 109,480.764 | 454.954 | <0.001 |
| | 2012–13–2011–12 | -1,185.242 | 1,820.500 | 0.524 |
| | 2013–14–2011–12 | -1,144.610 | 1,819.133 | 0.538 |
| | 2014–15–2011–12 | -4,244.181 | 1,819.716 | 0.033 |
| | 2015–16–2011–12 | 669.433 | 1,819.756 | 0.718 |
| | 2016–17–2011–12 | -1,653.588 | 1,818.975 | 0.377 |
| | 2017–18–2011–12 | -2,233.729 | 1,820.138 | 0.238 |
| | 2018–19–2011–12 | -1,771.906 | 1,820.500 | 0.345 |
| | **Random Effects** | **SD** | **Variance** | **ICC** |
| | Team | 2,130.215 | 4,537,816.447 | 0.234 |
| | Residual | 3,854.597 | 14,857,921.804 | |
| | Marginal $R^2$ / Conditional $R^2$ | | 0.090 / 0.303 | |
| All seasons | **Fixed Effects** | **Estimate** | **SE** | **p** |
| | Intercept | 109,885.411 | 216.473 | <0.001 |
| | Upper-Middle—Europe | 1,527.654 | 563.405 | 0.007 |
| | Lower-Middle—Europe | 600.019 | 537.319 | 0.266 |
| | Relegation—Europe | 169.865 | 657.800 | 0.797 |
| | **Random Effects** | **SD** | **Variance** | **ICC** |
| | Team | 2,556.585 | 6,536,128.037 | 0.332 |
| | Residual | 3,623.655 | 13,130,876.084 | |
| | Marginal $R^2$ / Conditional $R^2$ | | 0.018 / 0.344 | |

Note: SE is Standard Error; SD is Standard Deviation; ICC is Intraclass Correlation Coefficient. Statistical significance set at p<0.05.

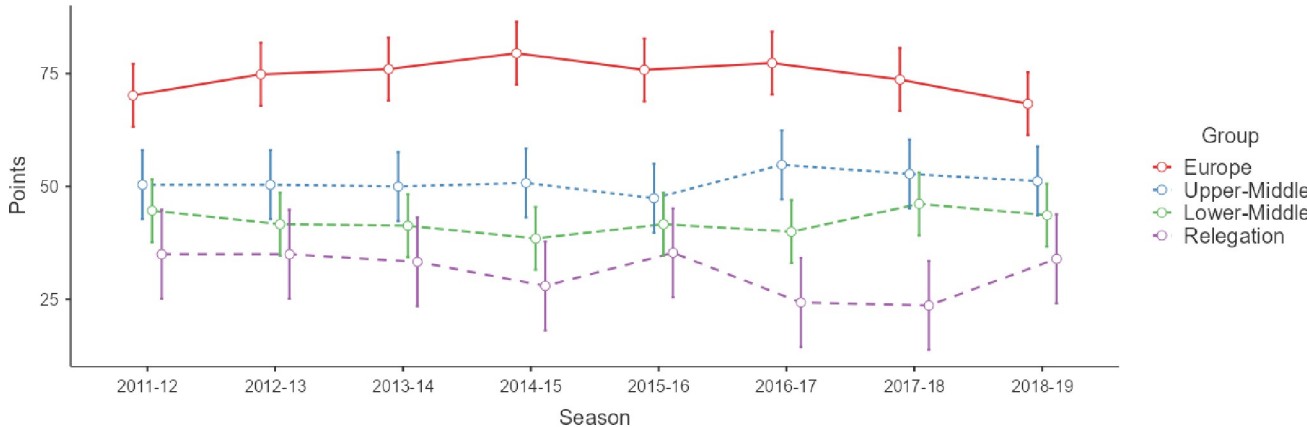

**Fig 1. Effects of season for each group and effects of group on the number of points accumulated.** Data represent the means and 95% confidence intervals.

p = 0.024), 2016–17 (-3,793.554; p = 0.010) and 2018–19 (-2,863.177; p = 0.047) compared to the 2011–12 season. In the Relegation group, the teams showed lower values of TD in 2014–15 (-4,244.181; p = 0.033) compared to the 2011–12 season. Likewise, Europe showed lower values of TD than Upper-Middle (-1,527.654; p = 0.007) during the whole period analysed.

Fig 1 shows the effects of season for each group and the effects of group on the number of points accumulated. In the Upper-Middle group, the teams showed more points in 2016–17 (5.018; p = 0.046) compared to the 2011–12 season. In the Lower-Middle group, the teams showed fewer points in 2014–15 (-5.855; p<0.001) and 2016–17 (-4.027; p = 0.022) compared to the 2011–12 season. In the Relegation group, the teams showed fewer points in 2016–17 (-10.667; p = 0.029) and 2017–18 (-11.333; p = 0.021) compared to the 2011–12 season. Likewise, Europe showed more points than Upper-Middle (13.474; p<0.001), Lower-Middle (21.795; p<0.001) and Relegation (32.177; p<0.001) during the whole period analysed.

## Discussion

The objective of this study was to analyse the performance of the Spanish *LaLiga* teams over a continuous period of eight seasons, considering the final league ranking. The main results of the study were that: 1) the Europe group showed significantly higher values compared to the other groups in most of the variables during the eight-season period; 2) the Europe group teams showed lower values of Length from the fifth season (from 2015–16 to 2018–19), and lower values of GkDef from the fourth season (from 2014–15 to 2018–19); 3) the Upper-Middle group teams showed lower values of Length from the fifth season (from 2015–16 to 2018–19); 4) the Lower-Middle group teams showed fewer Shots from the third season (from 2013–14 to 2018–19), and lower values of Length, GkDef and TD from the fourth season (from 2014–15 to 2018–19); and, 5) the Relegation group barely showed significant differences between seasons in any variable.

Regarding the Technical-Tactical dimension, the season factor had a significant effect on Crosses for Europe, Upper-Middle, Lower-Middle and Relegation, and a significant effect on Shots for Lower-Middle. The group factor also had a significant effect on Passes, Successful Passes and Shots. The distribution in these variables performed by the teams of the four groups implied greatly a performance stability throughout the analysed period. In relation to Passes and Successful Passes, the results of this work are similar to those of a recently published study about the evolution of physical and technical parameters in the Spanish *LaLiga* between the

2012–13 and 2019–20 seasons [12]. These researchers found no clear trend in the total passes as seasons progressed for any of the four groups but did find an upward trend in passing accuracy for the Top (ranked from 1st to 5th) and Lower-Middle (ranked from 11th to 15th) teams. However, the effect size of the differences between seasons was small. Therefore, it is worth mentioning that the study by Lago-Peñas et al. [12] also showed stability in the passes made during the analysed seasons. Bradley et al. [9], for their part, observed an increase in passes and successful passes made by the teams in the English *Premier League* over seven seasons (from 2006–07 to 2012–13). Tier A (teams ranked from 1st to 4th) and Tier C (teams ranked from 9th to 14th) teams significantly increased passes and successful passes made with a small effect size, Tier D (teams ranked from 15th to 20th) teams with a moderate effect size and Tier B (teams ranked from 5th to 8th) teams with a large effect size. A possible explanation for this could be that the teams located at the top of the ranking have been able to maintain a high and stable performance over the years, far from the more unstable performance of the rest of the teams located at the bottom of the ranking, whose annual objective is usually the one to maintain the category season after season. Another possible explanation could be that the technical-tactical dimension prevailed over the physical dimension throughout the seasons in the English *Premier League*. However, the results of the present work differ from those obtained by Bradley et al. [9].

With regard to the Crosses, it should be noted that the Europe teams showed fewer actions of this performance indicator in 2017–18 and 2018–19 compared to the 2011–12 season. Nevertheless, just like for the other three groups, the trend of Crosses over the eight seasons was quite stable for the Europe group. In the case of Shots, significant differences between seasons were only found for the Lower-Middle group. The teams of this group showed fewer Shots from the 2013–14 season. Lago-Peñas et al. [12], for their part, observed a significant decrease in the 2019–20 season compared to the 2012–13 season for the Top (from 1st to 5th) and Upper-Middle (from 6th to 10th) teams of the Spanish *LaLiga*. However, the effect size of these differences was small, and no trend was observed for any group as years passed. Therefore, the trend of the shots in the work of Lago-Peñas et al. [12] was quite stable throughout the period studied. When comparing the Technical-Tactical variables between groups throughout the period studied (the eight seasons together), Europe group obtained significantly higher values than the other three groups in Passes, Successful Passes and Shots. It seems that the frequency and effectiveness of shots and passes are some of the performance indicators that differentiate the most successful teams from the rest [21]. According to some works [22, 23], a high ball possession and, therefore, a high number of accumulated passes seem to be of great importance in the victory of football teams. In addition, a study that aimed to identify the statistics of the matches that best explain the success of football in the Spanish *LaLiga* using eight seasons as a sample (from 2010–11 to 2017–18), concluded that the two variables that best determine the success of a team are the effectiveness of the shots and the total number of shots made [24]. Therefore, the Europe group stood out for showing high values in the variables of the Technical-Tactical dimension that are most related to success.

With regard to the Set Piece dimension, the season factor had a significant effect on Corners and Fouls for Europe and Relegation, a significant effect on Goals for Upper-Middle, and a significant effect on Goals, Corners and Fouls for Lower-Middle. The group factor also had a significant effect on Goals, Corners and Fouls. The distribution in these variables performed by the teams of the four groups also represents a performance stability throughout the analysed period. It is worth noting that the Lower-Middle teams showed fewer Goals in 2013–14 and 2014–15 compared to the 2011–12 season. In these two seasons the teams of this group, in addition to showing fewer Shots, they showed less effectiveness in front of the rival goal. However, the trend of Goals over the eight seasons was quite stable for Lower-Middle. When

comparing the Set Piece variables between groups throughout the period studied, the Europe group showed significantly higher values than the other three groups in Goals and Corners. The key factor that can determine the result in a football match, and therefore the success of a team, is the goal. Castellano [25] found that the goals scored had a very high relationship with the achievement of a greater number of points at the end of the league competition in the Spanish *LaLiga* in the 2013–14 and 2014–15 seasons. It should also be noted that corner is a performance indicator related to attacking actions that, after the effectiveness of the shots and the total number of shots taken, can best determine the success of a team, since the action occurs near the rival goal [24]. A characteristic of the best-ranked teams in a league is that they often tend to get more set pieces such as corners after maintaining high ball possession [25], especially when possession occurs in the last third of the field, close to the opponent's goal [26]. Consequently, the success of the teams in the Europe group could be due to the fact that they also stood out for showing high values in variables that best explain the success of a team such as the goal and corner.

Regarding the Collective Tactical Behaviour dimension, the season factor had a significant effect on Length and GkDef for Europe, a significant effect on Length for Upper-Middle, and a significant effect on Length, Height and GkDef for Lower-Middle and Relegation. The group factor also had a significant effect on Width, Length, Height and GkDef. A significant decrease in Length values was found from the 2015–16 season for the Europe and Upper-Middle groups, and from the 2014–15 season for the Lower-Middle group. It seems that the teams of these groups increased the density of the effective playing space (same players in less space) as the seasons progressed. Furthermore, a significant decrease was found in GkDef values from the 2014–15 season for the Europe and Lower-Middle groups. This could be explained by the fact that the goalkeepers of these groups' teams are demanded to play a greater role in the offensive phase of the game, requiring his participation in initiating or continuing an attack with the players closest to him, such as with his centre-backs [10]. It could also be that these teams have been able to adopt a more defensive style of play due to less ball possession during matches. For its part, Relegation group showed a stable trend in this dimension over the eight seasons. Probably low values in the Collective Tactical Behaviour variables, represented in this group with low performance [25], may be one of the reasons that justify the stability in the collective behaviour described. When comparing the Collective Tactical Behaviour variables between groups throughout the period studied, the Europe group showed significantly higher values than the other groups in Width, Length, Height and GkDef. According to a previous study [25], a greater width, length and height of the defence was associated with the teams that accumulated the highest number of points at the end of the season in the Spanish *LaLiga* (in the 2013–14 and 2014–15 seasons). It seems, therefore, that the playing style of the most successful teams (e.g., higher positions in the final ranking) have higher values in the variables that represent the collective use of space as a trait.

In relation to the Physical dimension, the season factor had a significant effect on TD for Europe, Lower-Middle and Relegation. The group factor also had a significant effect on TD. Lower-Middle showed lower values of TD from the 2014–15 season. The teams in this group probably changed the way they played over the seasons, deploying lower total distance covered. However, the teams of the other three groups showed a stability in the total distance covered throughout the eight seasons. Lago-Peñas et al. [12] found a significant decrease in the total distance covered for different groups (Top, Upper-Middle, Lower-Middle and Lower) of the Spanish *LaLiga* over the eight seasons analysed (from 2012–13 to 2019–20). When comparing the Physical variable between groups throughout the period studied, Upper-Middle was the group that obtained the highest values in this physical variable, but it only showed significantly higher values than the Europe group. It is worth mentioning that some authors [27] indicate

that performance indicators of a technical-tactical nature have a greater influence than those of a conditional nature when determining the difference between the most successful teams in the championship. This is in line with the results presented by Castellano [25], who found that the total distance covered is not related to the success achieved by the teams (in this case of the Spanish men's top and second professional football division) at the end of the championship.

The trend in the number of points accumulated by the teams in the different groups of the Spanish *LaLiga* from 2011–12 to 2018–19 was stable. English authors [9] ensured that the teams in Tier A (from 1st to 4th) and Tier C (from 9th to 14th) groups of the *Premier League* accumulated, on average, 0.43 and 0.31 fewer points season after season (from 2006–07 to 2012–13), respectively, and for their part, the teams in Tier B (from 5th to 8th) and Tier D (from 15th to 20th) groups 0.32 and 0.20 more points, respectively. It seems that, throughout the seven seasons analysed by these researchers, the English teams in the Tier B group (from 5th to 8th) were closing the points gap with those that qualified for European competitions. However, this point difference between the English teams' season after season was minimal, so it is worth mentioning that the trend in the number of points accumulated in the English *Premier League* was also stable.

The main conclusion of the study is that the teams of the Europe, Upper-Middle and Relegation groups showed a quite stable performance, while the teams of the Lower-Middle group presented a worsening in different dimensions throughout the eight seasons analysed. It could be said that the Spanish football is in a plateau period in the performance of the best teams, which showed the ability to play in spaces with high player density as the seasons passed. Furthermore, they showed higher values in variables associated with success such as Passes, Success Passes, Shots and Corners, and in variables representative of the collective use of space (Width, Length, Height and GkDef) during the whole period studied. However, this does not detract from the fact that the teams that qualify in the less good half try to propose strategies that allow them in some cases to stay in the category, playing with the goalkeepers closer and closer to their defensive line. The information provided in the present study makes it possible to have reference values that have characterized the performance of the teams for each group.

The information provided in this study, especially due to the inclusion of a large volume of performances by the Spanish *LaLiga* teams (n = 5,518) over eight seasons, makes it possible to have reference values that have characterised the performance of the teams in the dimensions and variables studied based on league ranking at the end of each season. In addition, to the authors' knowledge, this is the first work to analyse the evolution of variables of the collective dimension (e.g., Width, Length, Height and GkDef) according to the final classification of the teams in a top-level football league over world level such as the Spanish *LaLiga*. However, the present study is not without limitations. Firstly, the performance of the teams was calculated using the means of the variables predefined by *Mediacoach*, without having the option of the authors' obtaining different variables by calculating them by accessing the raw data. Secondly, ball possessions were not considered in this study. The physical [28] and tactical [29] responses of the teams differ when the team has possession of the ball or not. This subject, distinguishing the attack and defence phase, is suggested for future research. Thirdly, the inclusion of other technical-tactical and physical variables (e.g., recoveries, duels, types of passes, accumulated distance at high-speed, number of accumulated sprints, etc.) and contextual variables such as the change of coach, the period of the season, playing at home or away or the level of the opponent [30–32], among others, could help refine possible inferences about the performance of the teams and to better explain their variability and stability over the years. Therefore, future studies should consider different technical-tactical and physical variables and different contextual variables. Finally, it should be noted that despite the fact that eight seasons in a national league (Spanish *LaLiga*) were studied in this study, caution must be taken when extrapolating

these league results to other countries or competitions with different characteristics [33]. Nevertheless, proposing this type of studies in other leagues or countries could help to better understand the evolution of the game on a more global level.

## Author Contributions

**Conceptualization:** Ibai Errekagorri, Julen Castellano.

**Data curation:** Ibai Errekagorri, Javier Fernandez-Navarro, Roberto López-Del Campo, Ricardo Resta, Julen Castellano.

**Formal analysis:** Ibai Errekagorri.

**Investigation:** Ibai Errekagorri, Javier Fernandez-Navarro, Julen Castellano.

**Methodology:** Ibai Errekagorri, Julen Castellano.

**Project administration:** Ibai Errekagorri, Julen Castellano.

**Resources:** Ibai Errekagorri, Javier Fernandez-Navarro, Roberto López-Del Campo, Ricardo Resta, Julen Castellano.

**Software:** Roberto López-Del Campo, Ricardo Resta.

**Supervision:** Javier Fernandez-Navarro, Roberto López-Del Campo, Julen Castellano.

**Validation:** Ibai Errekagorri, Javier Fernandez-Navarro, Roberto López-Del Campo, Ricardo Resta, Julen Castellano.

**Visualization:** Ibai Errekagorri, Julen Castellano.

**Writing – original draft:** Ibai Errekagorri.

**Writing – review & editing:** Ibai Errekagorri, Javier Fernandez-Navarro, Julen Castellano.

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
