## [Decision Letter · Decision Letter 0]

23 Jan 2024

PONE-D-23-35487An eight-season analysis of the teams' performance in the Spanish LaLiga according to the final league rankingPLOS ONE

Dear Dr. Errekagorri,

Thank you for submitting your manuscript to PLOS ONE. After careful consideration, we feel that it has merit but does not fully meet PLOS ONE’s publication criteria as it currently stands. Therefore, we invite you to submit a revised version of the manuscript that addresses the points raised during the review process.

We look forward to receiving your revised manuscript.

Kind regards,

Jovan Gardasevic

Academic Editor

PLOS ONE

Journal Requirements:

2. For studies involving third-party data, we encourage authors to share any data specific to their analyses that they can legally distribute. PLOS recognizes, however, that authors may be using third-party data they do not have the rights to share. When third-party data cannot be publicly shared, authors must provide all information necessary for interested researchers to apply to gain access to the data. (https://journals.plos.org/plosone/s/data-availability#loc-acceptable-data-access-restrictions) 

4. Please include your tables as part of your main manuscript and remove the individual files. Please note that supplementary tables (should remain/ be uploaded) as separate ""supporting information"" files

Additional Editor Comments:

I have read manuscript carefully, I feel that small corrections should be made to make it ready for publication. The manuscript is original, very methodologically well written, contains all the required parts, only minor corrections should be made in the Conclusion.

Reviewers' comments:

Reviewer's Responses to Questions

**Comments to the Author**

1. Is the manuscript technically sound, and do the data support the conclusions?

Reviewer #1: Yes

Reviewer #2: Yes

2. Has the statistical analysis been performed appropriately and rigorously? 

Reviewer #1: Yes

Reviewer #2: Yes

3. Have the authors made all data underlying the findings in their manuscript fully available?

Reviewer #1: Yes

Reviewer #2: No

4. Is the manuscript presented in an intelligible fashion and written in standard English?

Reviewer #1: Yes

Reviewer #2: Yes

5. Review Comments to the Author

Reviewer #1: I think the topic is interesting.

The idea of finding elements that bring advantage is useful for practice.

The manuscript is technically adequately prepared.

The authors have my support. I recommend that the manuscript be published in its current form.

Reviewer #2: Thank you very much for giving me a chance to read the study under the title “An eight-season analysis of the teams' performance in the Spanish LaLiga according to the final league ranking” (PONE-D-23-35487). I have carefully read the manuscript and my opinion is that the manuscript has a merit to be published in your reputable journal with some minor corrections. The authors aimed to analyze the Spanish LaLiga teams' performance taking some key competitive performance variables into account over a continuous period of eight seasons according to the final league ranking. The manuscript is original, informative and readable while the study design and writing style are on the adequate level. The introduction is well written, as well as materials and methods section that is very well prepared and organized according to contemporary methodological rules. At the end, I have no amendments on results and discussion part but I would recommend to the authors to prepare the conclusion part in the following order: the main conclusions, the limitations of the study (more precisely) as well as recommendations for the further studies (it is very important to briefly elaborate it and highlight the most important notes) as this part might help some upcoming researchers to set up the monitoring systems to suit them more in the future. Lastly, I would recommend to accept this manuscript right after the authors revise it in the adequate manner.

6. PLOS authors have the option to publish the peer review history of their article (what does this mean?). If published, this will include your full peer review and any attached files.

Reviewer #1: No

Reviewer #2: **Yes: **Stevo Popovic

---

## [Author Response · Author response to Decision Letter 0]

3 Feb 2024

Dear Jovan Gardasevic,

Please find enclosed the resubmission of our article entitled “An eight-season analysis of the teams' performance in the Spanish LaLiga according to the final league ranking” (manuscript ID PONE-D-23-35487). We thank a lot the reviewers’ comments. We have answered the points raised by the Reviewer #2 and we feel they have allowed us to improve our work substantially. All the changes included in the manuscript are with track changes. 

Sincerely,

Ibai Errekagorri

• Reviewer #1:

­ Comment: I think the topic is interesting. The idea of finding elements that bring advantage is useful for practice. The manuscript is technically adequately prepared. The authors have my support. I recommend that the manuscript be published in its current form.

­ Answer: Thanks a lot for your words and for your recommendation to publish the article. We are very glad that you liked the work.

• Reviewer #2:

­ Comment: Thank you very much for giving me a chance to read the study under the title “An eight-season analysis of the teams' performance in the Spanish LaLiga according to the final league ranking” (PONE-D-23-35487). I have carefully read the manuscript and my opinion is that the manuscript has a merit to be published in your reputable journal with some minor corrections. The authors aimed to analyze the Spanish LaLiga teams' performance taking some key competitive performance variables into account over a continuous period of eight seasons according to the final league ranking. The manuscript is original, informative and readable while the study design and writing style are on the adequate level. The introduction is well written, as well as materials and methods section that is very well prepared and organized according to contemporary methodological rules. At the end, I have no amendments on results and discussion part but I would recommend to the authors to prepare the conclusion part in the following order: the main conclusions, the limitations of the study (more precisely) as well as recommendations for the further studies (it is very important to briefly elaborate it and highlight the most important notes) as this part might help some upcoming researchers to set up the monitoring systems to suit them more in the future. Lastly, I would recommend to accept this manuscript right after the authors revise it in the adequate manner.

­ Answer: Thanks a lot for your words and for your recommendation. We have modified the conclusions section according to your suggestions.

---

## [Editor Report · Decision Letter 1]

8 Feb 2024

An eight-season analysis of the teams' performance in the Spanish LaLiga according to the final league ranking

PONE-D-23-35487R1

Dear Mr. Ibai Errekagorri,

We’re pleased to inform you that your manuscript has been judged scientifically suitable for publication and will be formally accepted for publication once it meets all outstanding technical requirements.

Kind regards,

Jovan Gardasevic

Academic Editor

PLOS ONE

---

## [Editor Report · Acceptance letter]

15 Feb 2024

PONE-D-23-35487R1 

PLOS ONE

Dear Dr. Errekagorri, 

I'm pleased to inform you that your manuscript has been deemed suitable for publication in PLOS ONE. Congratulations! Your manuscript is now being handed over to our production team.

Kind regards, 

on behalf of

Dr. Jovan Gardasevic 

Academic Editor

PLOS ONE